# LANGUAGE SPECIFIC KNOWLEDGE: DO MODELS KNOW BETTER IN *X* THAN IN ENGLISH?

## ABSTRACT

Often, multilingual language models are trained with the objective to map semantically similar content (in different languages) in the same latent space. In this paper, we show a nuance in this training objective, and find that *by changing the language of the input query, we can improve the question answering ability of language models*. Our contributions are two-fold. First, we introduce the term **Language Specific Knowledge (LSK)** to denote queries that are best answered in an "expert language" for a given LLM, thereby enhancing its question-answering ability. We introduce the problem of language selection—for some queries, language models can perform better when queried in languages other than English, sometimes even better in low-resource languages—and *the goal is to select the optimal language for the query*. Second, we introduce simple to strong baselines to test this problem. Additionally, as a first-pass solution to this novel problem, we design **LSKEXTRACTOR** to benchmark the language-specific knowledge present in a language model and then exploit it during inference. To test our framework, we employ three datasets that contain knowledge about both cultural and social behavioral norms. Overall, LSKEXTRACTOR achieves up to 10% relative improvement across datasets, and is competitive against strong baselines, while being feasible in real-world settings. Broadly, our research contributes to the open-source[1] development of language models that are inclusive and more aligned with the cultural and linguistic contexts in which they are deployed.

## 1 INTRODUCTION

Language models are trained to understand and generate responses in dozens of languages, and are trained with either monolingual or parallelly translated data (Singh et al., 2024). Multilingual language models are trained so that two sentences that are semantically similar but in different languages are mapped to the same point in the latent space (Xu et al., 2025; Gurgurov et al., 2024; Pfeiffer et al., 2022; Ruder et al., 2019) (what we coin as the **"latent language alignment hypothesis"**). This hypothesis applies to sentences in all languages, creating multilingual language models. This hypothesis is supported by current reports on DeepSeek-R1 (DeepSeek-AI et al., 2025) spontaneously switching to Chinese during its chain-of-thought, even when presented with an English query (Marjanović et al., 2025). However, the same hypothesis has been challenged by works like the Multilingual Trolley Problem (Jin et al., 2025) which show that the alignment of multilingual language models to human preferences varies with the language of the input query.

Figure 1 presents another case in which the hypothesis of latent language alignment does not hold. In this example, we ask Llama-3.1-8B-Instruct about the sport that American women tend to watch the most in different languages (see the caption for details of this toy experiment). The model produces different answers across languages, with only Hindi and Japanese yielding correct responses. If the languages were truly aligned in the latent space, we would expect the model to produce the same output regardless of the input language. This inconsistency highlights limitations of the latent language alignment hypothesis, which arise from known sources of cross-lingual misalignment such as non-compositionality (the meaning of a phrase cannot be deduced directly from the individual words, i.e., metaphors and idioms (Sathe et al., 2024; Cheng and Bhat, 2024; Zhou et al., 2023)) and non-isomorphism (words lacking direct translations (Wu et al., 2024)). Building on this perspective,

---

*These authors contributed equally to this work.
[1]https://anonymous.4open.science/r/LSKExtractor-272F/

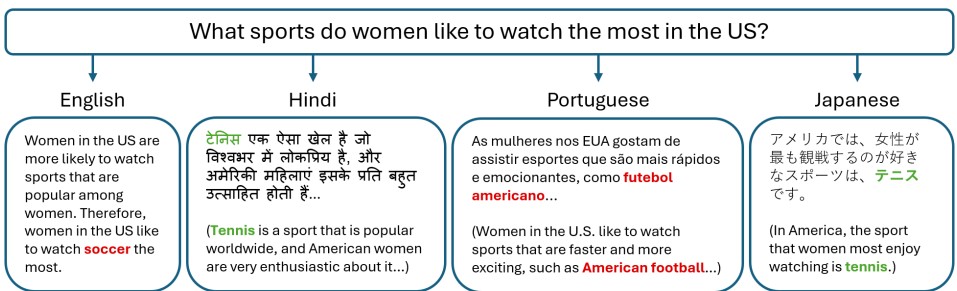

Figure 1: In this toy experiment, we prompt Llama-3.1-8B-Instruct with the same question across multiple languages (shown in English here only for illustration; the actual queries were translated into each respective language). The correct answer is **tennis**, yet the model produces different outputs depending on the query language. This illustrates what we refer to as Language-Specific Knowledge.

we propose another source of misalignment—this time within language models themselves—which we call Language Specific Knowledge.

We define **Language Specific Knowledge (LSK)** as knowledge that a language model appears to access more readily or represent more accurately when queried/asked to reason in a particular language (the *expert language*). In Figure 1, the varying responses across languages are evidence of LSK. Rather than viewing this as a limitation, we argue that such behavior should be leveraged in a more informed and intentional manner, allowing us to guide language models toward languages that may yield more accurate, aligned, and culturally appropriate responses for a given topic.

We propose a novel, two-stage approach, called **LSKEXTRACTOR** (see Figure 2), that is designed to identify the expert language for a particular query and model, for the most accurate answer. In the first stage of "Mapping LSK", we map out LSK and their corresponding expert languages by conducting chain-of-thought (CoT) reasoning in 16 languages on training queries from three datasets (CultureAtlas (Fung et al., 2024), BLEnD (Myung et al., 2025), and Social IQa (Sap et al., 2019)). We use language-specific reasoning to ensure the model is using the knowledge embedded within the corresponding language. These queries are clustered in a shared semantic space, and each cluster is assigned an expert language based on the CoT language that achieves the highest performance within that group. In the second stage of "LSK-Informed Reasoning", during test-time inference, we embed an unseen query into the same space to identify its corresponding cluster and retrieve the optimal language(s) for reasoning. The final answers are generated using CoT in the language identified as the expert for that knowledge region. This scalable method allows models to draw upon multilingual strengths dynamically, relatively improving accuracy by 10% across datasets without additional fine-tuning across all models and datasets. Our method is comparable to strong baselines (that we also propose) while still providing a feasible solution applicable in real world settings.

Our contributions are as follows:

- We formally define **Language Specific Knowledge (LSK)** and provide intuitive and empirical evidence of its presence in multilingual language models.
- We propose **LSKEXTRACTOR**, a scalable two-stage framework that identifies expert languages for specific knowledge regions and leverages this LSK-to-language map to improve inference through strategically switching the query language.
- Finally, we conduct systematic experiments across multiple state-of-the-art models to evaluate the effects of language-specific reasoning performance across topics and inform the benefits of LSKEXTRACTOR.

## 2 RELATED WORK

Prior work has examined how language influences model reasoning (Schut et al., 2025; Zhong et al., 2024; Yong et al., 2025), effects of language on model alignment with human preferences (Jin et al., 2025; Durmus et al., 2024), and cross-linguistic generalization (Chang et al., 2022). Chang et al. (2022) investigated how different languages are represented within the XLM-R multilingual model. They found that languages occupy distinct regions in the representational space, though languages

with similar distributions can be aligned through mean-shifting. This indicates that semantically equivalent sentences in different languages may not map to the same low-level representations. This insight motivates our study by highlighting the need for language-specific knowledge representations when reasoning or answering questions across linguistic boundaries.

Other works have focused specifically on multilingual reasoning. For instance, Schut et al. (2025) demonstrated that language models tend to default to English during internal reasoning, which can negatively impact downstream task performance, fluency, and fairness. We extended this finding by identifying, for some given topic, the language in which a multilingual language model exhibited greater expertise. Similarly, Zhong et al. (2024) found that models often reason internally in a specific language and exhibit cultural biases aligned with that language when responding to culturally grounded questions. In our work, we aim to boost multilingual reasoning by identifying such LSK and strategically leveraging expert languages where such knowledge is most richly encoded through the LSKEXTRACTOR. This complements approaches like Huang et al. (2024) that merge external multilingual representations to enhance general understanding, Ziabari et al. (2025), which adapt LLM reasoning between intuitive (System 1) and deliberative (System 2) modes based on task needs, or even Huang et al. (2023) that encourages language models to think in other languages to improve performance. We adapt this as a baseline, called the LLMSelected baseline.

Several works have investigated multilingual reasoning from different perspectives: improving reasoning in low-resource languages (Senel et al., 2024), benchmarking the reasoning abilities of language models across languages (Etxaniz et al., 2023; Kumar et al., 2025; Gao et al., 2025), and enhancing semantic alignment between languages (Yoon et al., 2024). These efforts primarily aim to strengthen cross-lingual semantic representations to support more consistent reasoning across languages. In a related line of work, Yong et al. (2025) demonstrated that chain-of-thought traces in various languages can be aligned to their English counterparts to facilitate multilingual reasoning. In contrast, we highlight a fundamental limitation of this alignment approach: certain languages encode concepts that do not have direct equivalents in others. This observation underscores the lack of a universal one-to-one mapping across languages (Liu et al., 2024). **Rather than enforcing alignment, our work embraces linguistic diversity by leveraging the unique conceptual affordances of each language to enhance reasoning performance.**

Furthermore, language is an important part of model alignment with human preferences. However, prior work has shown that current multilingual models are not well aligned with humans, showing more US and Euro-centric representations rather than multicultural (Durmus et al., 2024; Rystrøm et al., 2025). Recent studies have shown that languages are indeed proxies for culture Adilazuarda et al. (2024), thus they should be aligned to culturally diverse preferences. However, even when prompted across different languages, they fail to align with these culturally diverse moral preferences (Jin et al., 2025). Our work contributes to alignment by identifying the expert language for specific domains of knowledge and demonstrating how strategically using these languages can elicit responses that better reflect localized, culturally grounded human preferences.

## 3 LSKEXTRACTOR METHODOLOGY

Given an LLM, LSKEXTRACTOR aims to identify the most effective language for answering an LSK question. We first define a set of candidate languages $\mathcal{L}$. For a specific language $l \in \mathcal{L}$, let $Q_l$ denote the query $Q$—consisting of the question together with the model instruction—translated into $l$. We then denote the performance of the (multilingual) language model $LLM_\theta$ (with parameters $\theta$) on $Q_l$ from a dataset with CoT reasoning in $l$ as $Acc(LLM_\theta(Q_l \mid l))$. We also compute the performance of the model without CoT reasoning, denoted simply as $Acc(LLM_\theta(Q_l))$. The complete set of model prompts is provided in Appendix H, and the query translation process is detailed in Appendix D.

We use this formulation to map LSK to an expert language, cluster semantically similar queries, and form a language-topic alignment map (as detailed in the next paragraph). We can, then, take advantage of the language-topic alignment map during testing by identifying the topic cluster and using the corresponding language for reasoning. Figure 2 contains an overview of our solution. We detail the steps in each stage below.

In the first stage of LSKEXTRACTOR, we construct an LSK-to-language mapping for each model $LLM_\theta$ and dataset $D$:

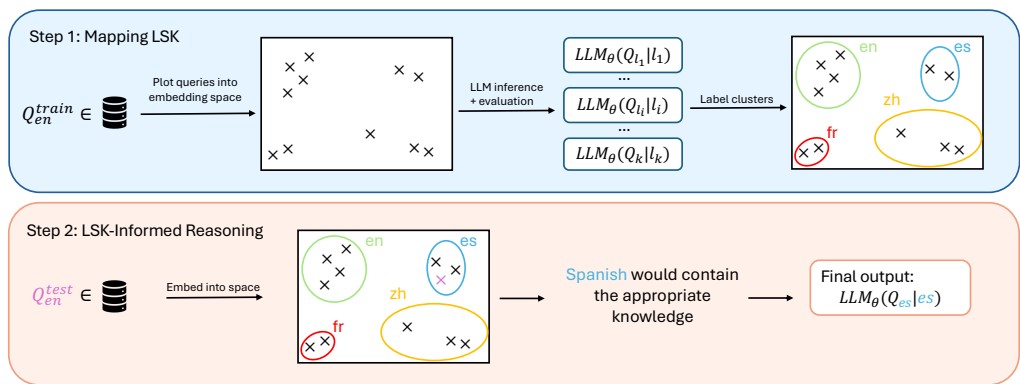

Figure 2: Overview of LSKExtractor. Our method consists of two main steps. In Step 1, we embed training queries into a shared semantic space and cluster them based on topical similarity. For each cluster, we determine the expert language—i.e., the language that yields the most accurate or contextually appropriate reasoning—by comparing model responses across languages. In Step 2, during test-time inference, we embed the test query into the same space, identify its nearest cluster, and select the corresponding expert language (e.g., Spanish) to guide the model toward producing a more informed and culturally grounded response.

1. For each training query $Q^{\text{train}}$, embed the English version $Q^{\text{train}}_{\text{en}}$ using an embedding model[2], and cluster the embeddings into $k$ groups using the $k$-means algorithm. Let the resulting clusters be $C_1, C_2, \ldots, C_k$.

2. For each query $Q \in C_j$ and each candidate language $\ell \in \mathcal{L}$, evaluate the model's performance with CoT reasoning, $Acc(LLM_\theta(Q_l \mid \ell))$.

3. For each cluster $C_j$, compute the average accuracy of $LLM_\theta$ for the queries in $C_j$ when the model is asked to reason in $\ell$ (denoted as $Acc_\ell(C_j)$):
$Acc_\ell(C_j) = \frac{1}{|C_j|} \sum_{Q \in C_j} Acc(LLM_\theta(Q_l \mid \ell))$

4. Assign to each cluster $C_j$ its expert language by selecting the maximizer:
$\ell^*(C_j) = \arg\max_{\ell \in \mathcal{L}} Acc_\ell(C_j)$

In the second stage, we leverage the LSK representation to guide test-time inference (to clarify, $Q^{\text{test}}_{\text{en}}$ is a test query in English):

1. Embed each test query $Q^{\text{test}}_{\text{en}}$ into the same semantic space as used during training.

2. Assign $Q^{\text{test}}_{\text{en}}$ to its nearest cluster $C_j$ based on embedding cosine similarity.

3. Retrieve the expert language $\ell^*(C_j)$ assigned to cluster $C_j$.

4. Perform CoT reasoning in this expert language: $LLM_\theta(Q_{\ell^*(C_j)} \mid \ell^*(C_j))$ and use the result as the final model output.

## 4 EXPERIMENTS

### 4.1 EXPERIMENTAL SETUP

**Languages.** For our experiments, we set $\mathcal{L}$ to include the following 16 languages: Arabic, Bengali, Chinese, English, French, German, Hindi, Italian, Japanese, Korean, Portuguese, Russian, Spanish, Thai, Turkish, and Vietnamese. It is important to note that $\mathcal{L}$ is treated as a hyperparameter: the methodology selects the best expert language from the available set. Crucially, a model's multilingual coverage does not need to align with the chosen set of languages, since LSKExtractor is designed to guide language selection based on performance observed in the training samples.

---

[2]We use the QWEN/QWEN3-EMBEDDING-0.6B model due to its strong performance on the MTEB (Muennighoff et al., 2022; Enevoldsen et al., 2025) and lightweight nature.

**Datasets.** We hypothesize that language-specific knowledge may manifest in culture, societal norms, and common sense reasoning. Math, coding, and logic are examples of domains that we expect to have little LSK, which is why we do not evaluate on those domains. Hence, we select three datasets that reflect these properties:

- **CultureAtlas** (Fung et al., 2024): a dataset consisting of cultural norms (e.g., "During the Chinese New Year, in Southern China, red envelopes are typically given by the married to the unmarried[...]"), labeled as either True or False. To create a more challenging task, we reformat the dataset into multiple-choice questions (MCQs) with four answer options: one true claim and three false claims about the same country. Further details are provided in Appendix F.
- **BLEnD** (Myung et al., 2025): a multiple-choice question answering dataset where the input is a societal norm (e.g., "What is the common dress code for school teachers in Azerbaijan?") and four answer choices (e.g., "A. apron, B. black formal suit, C. uniform, D. shirt"). The output is one of the selected answer choices.
- **Social IQa** (Sap et al., 2019): a multiple-choice common sense reasoning dataset where the input contains some context (e.g., "Sydney walked past a homeless woman asking for change but did not have any money [...] Sydney felt bad"), a question (e.g., "How would you describe Sydney?"), and three answer choices (e.g., "A. sympathetic, B. like a person who was unable to help, C. incredulous"). The output is one of the answer choices.

We use 8k instances for training and 2k for testing on BLEnD and Social IQa datasets. For CultureAtlas, due to reformatting, we use 5k instances for training and 1.5k for testing. Since all datasets are framed as classification tasks, **we measure and report performance with classification accuracy** ((# of True Positive + # of True Negative) / # of All Predictions). Although we focus on classification tasks, we can easily extend this to generation tasks where accuracy is measured by applying a threshold to a response quality metric (e.g., $> 0.7$ ROUGE score is accurate and $< 0.7$ ROUGE score is inaccurate).

**Models.** For our evaluation, we use a variety of model sizes from a variety of families: Google's `gemma-3-1b-it` and `gemma-3-12b-it` (Team et al., 2024), Meta's `Llama-3.2-1B-Instruct`, `Llama-3.2-3B-Instruct`, and `Llama-3.1-8B-Instruct` (Grattafiori et al., 2024), Qwen's `Qwen3-0.6B`, `Qwen3-8B`, and `Qwen3-14B` (Yang et al., 2025), and Cohere-Lab's `aya-23-8B` (Dang et al., 2024). We use instruction-tuned versions because those models are trained to handle multilingual inputs.

**Methods.** To better understand model performance on these datasets, and to highlight the advantages of selecting the most informative expert language, we compare LSKEXTRACTOR against several baseline methods:

**(1) Simple Baseline.** The simplest approach evaluates the model only in English, the original data language. This provides a reference for base model performance with and without explicit reasoning:

- **Only English**: Base performance of the models in English (the original data language), with and without reasoning: $LLM_\theta(Q_{\text{en}} \mid \text{en})$ and $LLM_\theta(Q_{\text{en}})$,

**(2) Simple LSK Baselines.** These methods test the hypothesis that languages other than English can be more informative (i.e., demonstrate the existence of LSK). Importantly, they do not rely on additional assumptions such as country or cultural labels:

- **LLM-Selected**: In order to test whether a language model has an internal LSK-to-language mapping captured by its weights, at test time, $LLM_\theta$ is given $Q_{\text{en}}$ and is asked to select the most appropriate language $\ell \in \mathcal{L}$ in which to answer the query. Then, we use $LLM_\theta(Q_l \mid l)$ and $LLM_\theta(Q_l)$ for evaluation. Prompt for language selection details are provided in Appendix E, Figure 13,
- **Best Global Language**: Performance with the best-performing language $x \in \mathcal{L}$, with and without reasoning: $LLM_\theta(Q_x \mid x)$ and $LLM_\theta(Q_x)$, where $x = \arg\max_{\ell \in \mathcal{L}} Acc(LLM_\theta(Q_\ell \mid \ell))$ on the training set of a particular dataset,

**(3) Strong LSK Baselines.** These methods also leverage the presence of LSK, but they make stronger assumptions about the data or are computationally less feasible in real-world scenarios:

- **Majority Voting**: Performance using majority voting across all languages $\ell \in \mathcal{L}$, with and without reasoning: $\text{MajorityVote}\big(\{Acc(LLM_\theta(Q_\ell \mid \ell))\}_{\ell \in \mathcal{L}}\big)$ and $\text{MajorityVote}\big(\{Acc(LLM_\theta(Q_\ell))\}_{\ell \in \mathcal{L}}\big)$,
- **Country Mapping**: At test time, we also evaluate a setting where the query language $\ell$ is chosen based on a country–language mapping according to the most spoken language in the region (e.g., Hindi for India). This setting applies only to CultureAtlas and BLeND, which include country labels for each question. Details of the country–language mapping are provided in Appendix G.

## 4.2 RESULTS

In this section, we present empirical results comparing LSKEXTRACTOR against a diverse set of baselines and datasets. Our analysis is structured around the following research questions:

- **RQ1:** How well does LSKEXTRACTOR perform relative to both simple and strong baselines, and under what conditions does it provide the largest gains?

- **RQ2:** How does the clustering component influence performance within LSKEXTRACTOR, and does grouping queries into semantically coherent clusters lead to more effective language selection?

- **RQ3:** Which languages are chosen in different clusters, and what does this reveal about the presence of LSK and its utility for question answering?

- **RQ4:** Can the LSK maps learned during Step 1 of LSKEXTRACTOR be transferred across models and datasets, and if so, why?

Together, these questions guide our evaluation of LSKEXTRACTOR, enabling us to assess not only its raw effectiveness but also its feasibility, interpretability, and the underlying dynamics of multilingual knowledge access.

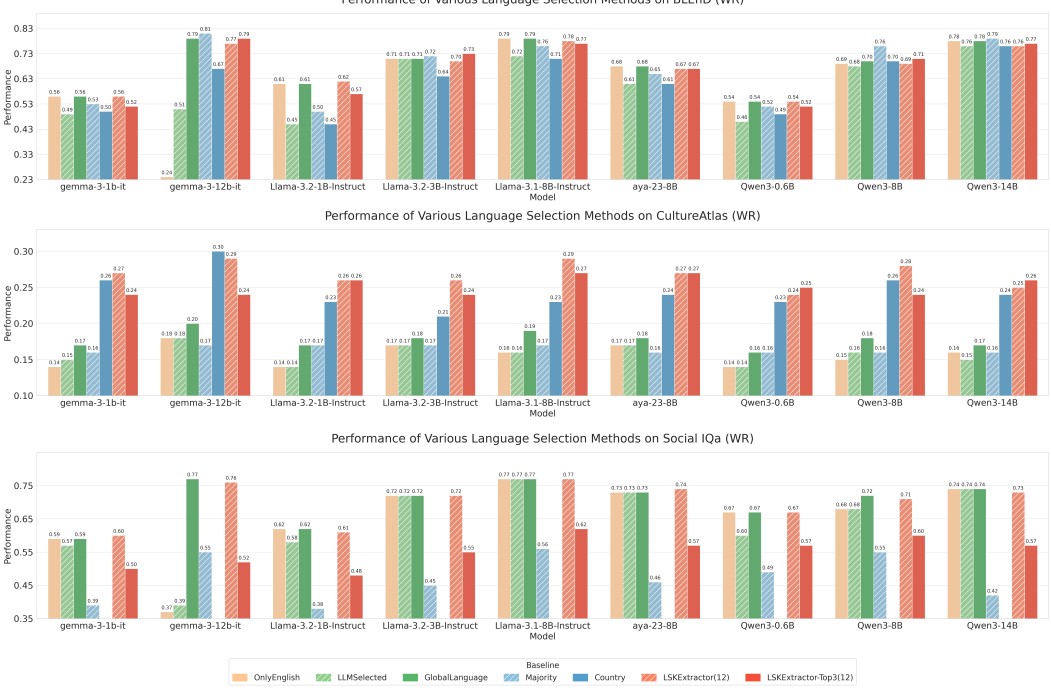

Figure 3: The main results of measuring LSK – we show the performance of our various baselines and LSK across the three datasets. This setting is *with reasoning*, as opposed to Figure 10 in Appendix C.

### 4.2.1 RQ1: COMPARATIVE PERFORMANCE OF LSKEXTRACTOR

Figure 3 presents the performance of LSKEXTRACTOR and our baselines (Section 4.1) across all three datasets. We also include LSKEXTRACTOR-Top3, a variant of LSKEXTRACTOR that applies majority voting across the top three languages within each cluster. Overall, the results demonstrate that LSKEXTRACTOR consistently outperforms or matches the strongest baselines. On the most challenging dataset, CultureAtlas, LSKEXTRACTOR achieves a 10.4% relative improvement over the best-performing baseline. On BLEnD, LSKEXTRACTOR improves on OnlyEnglish by an average of 23.6%, achieving performance comparable to GlobalLanguage. On Social IQa, LSKEXTRACTOR provides an average improvement of 11.9% over OnlyEnglish, again reaching performance on par with GlobalLanguage. Importantly, **LSKEXTRACTOR also outperforms the computationally expensive Majority Voting baseline across all datasets** (BLEnD: +1.8%, CultureAtlas: +63.4%, Social IQa: +49.7%), indicating that many of the languages included in evaluation are suboptimal and cannot reliably serve as expert languages for these queries.

For the LLMSelected baseline, we observe that models often default to English (Appendix E, Figure 12), which explains why the performance of LLMSelected closely matches that of OnlyEnglish in Figure 3. Exceptions highlight the risks of relying on the model's internal mapping. For example, Qwen3-0.6B on Social IQa selects Chinese as its preferred language (Figure 12), leading to lower accuracy (60%) than OnlyEnglish (67%) in Figure 3. Conversely, gemma-3-12b-it on BLEnD selects a diverse set of languages and achieves substantially higher accuracy (51%) compared to OnlyEnglish (24%). These findings suggest that **language models cannot yet reliably articulate their internal LSK-to-language mapping**, even when such mappings clearly exist in their learned representations.

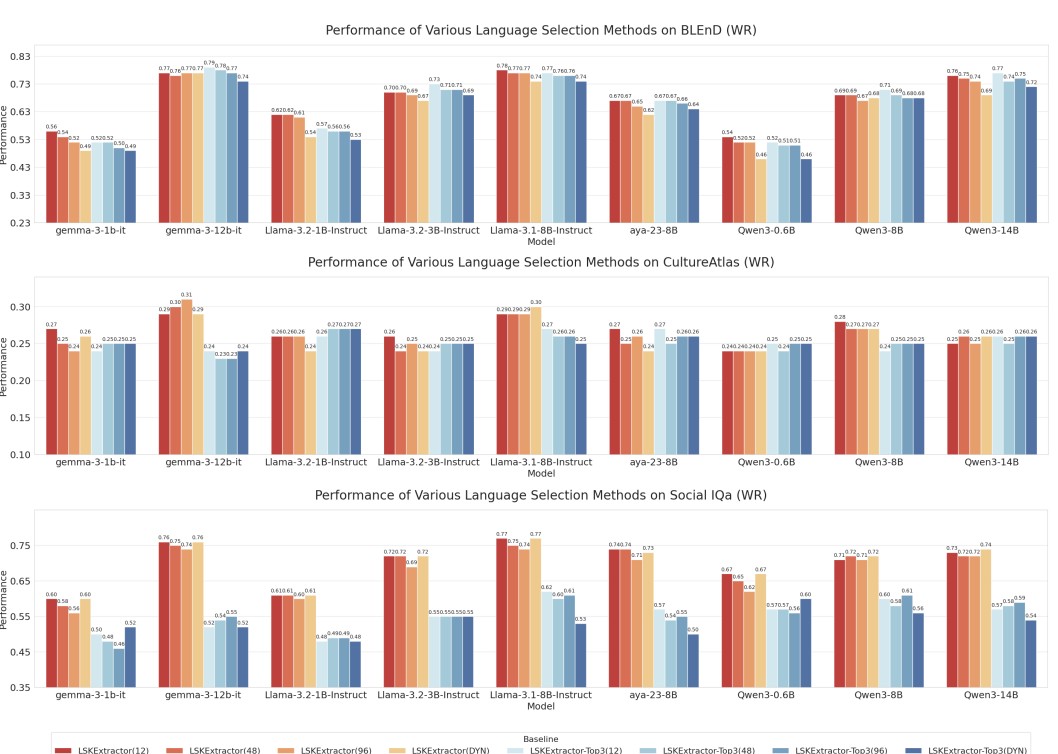

Figure 4: Understanding the impact of the clustering on LSKExtractor with 12, 49, and 96 clusters using the kmeans++ algorithm, and the HDBSCAN method (labeled as "DYN").

### 4.2.2 RQ2: CLUSTER SIZE VERSUS PERFORMANCE

Figure 4 reports the performance of LSKEXTRACTOR and LSKEXTRACTOR-Top3 under different clustering configurations. We vary the number of clusters in $k$-means (12, 48, and 96) and also include results from HDBSCAN, a dynamic clustering algorithm denoted as "DYN" in the figure.

Overall, the choice of clustering method and cluster size has only a modest effect on performance. However, **we observe a general trend of decreasing accuracy as the number of clusters increases**.

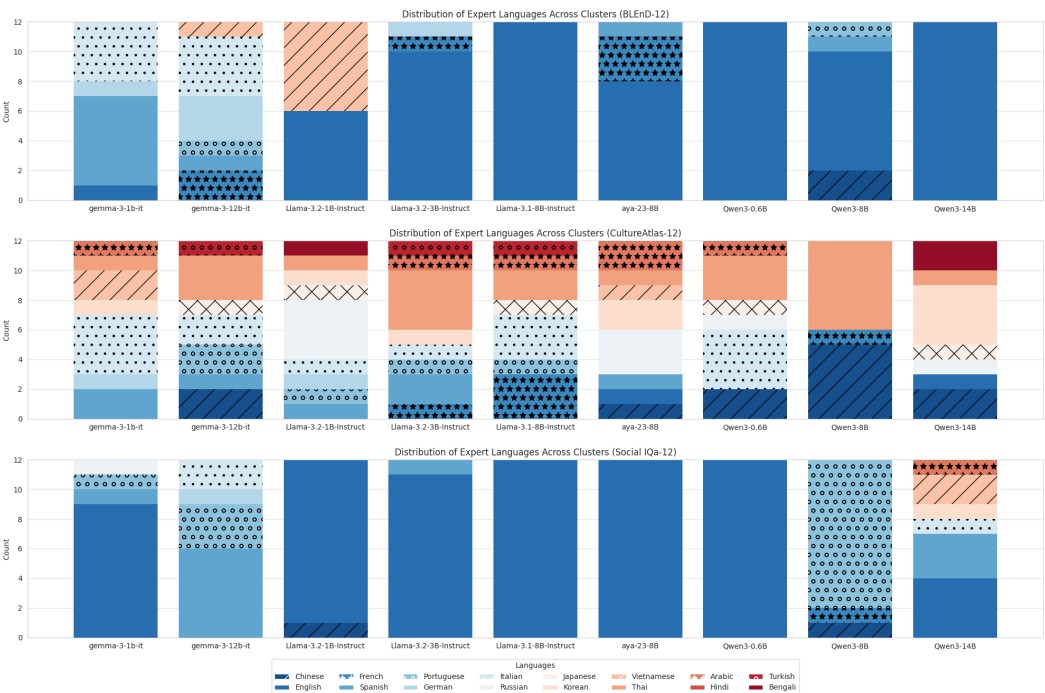

Figure 5: Distribution of languages selected across clusters (12-means clustering), across datasets.

### 4.2.3  RQ3: LANGUAGES IN EACH CLUSTER

Figure 5 shows stacked bar plots of the languages selected by LSKEXTRACTOR across clusters (with corresponding plots for other clustering methods in Figures 7–9 in Appendix B). When paired with the performance results in Figure 3, several clear patterns emerge. For both BLEnD and Social IQa, English dominates as the selected language across most clusters. This aligns with their baselines: OnlyEnglish and GlobalLanguage perform strongly on these datasets, and LSKEXTRACTOR similarly selects English in most cases, resulting in comparable performance. Still, we observe interesting deviations. For example, Llama-3.2-1B selects Vietnamese in roughly half of the clusters for BLEnD, while Qwen3-8B selects Portuguese in nearly 80% of the clusters for Social IQa. These cases highlight that **certain models may exhibit biases toward particular non-English languages, even when English is globally optimal**. The dominance of English in Social IQa is unsurprising: the dataset consists of commonsense reasoning questions rooted in Western traditions, which (1) makes English the best-performing GlobalLanguage, (2) drives LLMs to favor English when selecting expert languages, and (3) explains why OnlyEnglish already achieves strong results. By contrast, CultureAtlas presents a much more diverse distribution of languages across clusters, reflecting the dataset's cultural and region-specific grounding. In this setting, LSKEXTRACTOR consistently outperforms the baselines (Figure 3), underscoring its advantage in identifying the most informative language for each query. We observe that different models often select different expert languages for the same cluster. This is especially interesting for Culture Atlas clusters, which are—as shown in our cluster-level analysis (Appendix B)—aligned with specific countries. Taken together, these findings show that **LSKEXTRACTOR adapts flexibly to dataset characteristics: converging on English when it is globally optimal, while diversifying language selection when LSK is present**.

### 4.2.4  RQ4: TRANSFERABILITY OF LSK-MAP ACROSS DATASETS AND MODELS.

We test the robustness of LSKEXTRACTOR's LSK-to-language map. We run two experiments to test the transferability of the LSK map across models and datasets: (1) "transfer-model": we use the Llama-3.1-8B-Instruct's CultureAtlas LSK map to evaluate a different model on CultureAtlas. (2)

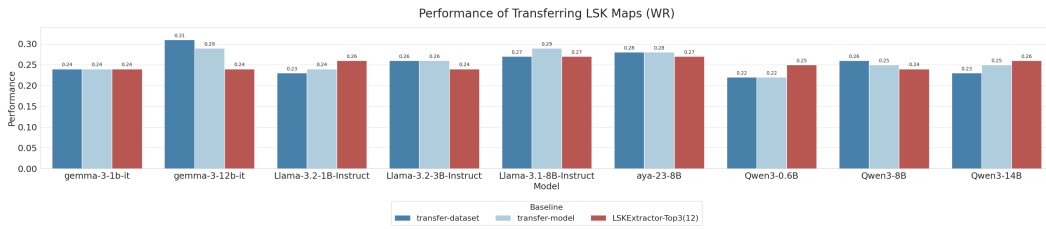

Figure 6: Performance of language models using LSKEXTRACTOR when the LSK map is transferred from another setting. "transfer-model" is when the LSK map of Llama-3.1-8B-Instruct is used to evaluate the performance on another LLM. "transfer-dataset" is when the LSK map for BLEnD's training set is used to evaluate the same model on CultureAtlas's testing set.

"transfer-dataset": we use a model's BLEnD LSK map to evaluate the same model on CultureAtlas. Figure 6 shows the results: **LSK maps can be quite robust to dataset and model**. The former can be explained by the fact that the LSK map's reliance on semantic content causes the map to be transferrable across datasets – the same kinds of queries would be best performed in a particular language. The latter can be explained by the overlap across the chosen expert languages themselves. According to Figure 5, there are some overlaps in the chosen expert languages across clusters. We also want to point out that choosing one expert language does not mean other expert languages do not exist. In our experiments, we noticed multiple languages having the same $Acc(C_j)$ as the expert language, and require a tie-breaker.

### 4.3 FEASIBILITY ADVANTAGES OF LSKEXTRACTOR

Overall, we observe that LSKEXTRACTOR performs on par with, if not better than, the strongest baseline methods, while offering significant advantages in terms of feasibility. Unlike the Country-to-Language mapping approach, which relies on a simplistic heuristic of assigning countries to their most spoken languages, LSKEXTRACTOR does not require explicit labeling of country information to guide language selection. In real-world scenarios where queries may involve complex entities, span multiple cultural contexts, or lack clear country associations, obtaining such labels is often impractical or infeasible.

Another competitive baseline is the majority-vote method. While conceptually straightforward, this approach is prohibitively expensive, as it requires querying across all available languages. Moreover, it implicitly assumes that all languages are equally informative. However, as demonstrated in our examples, this assumption is flawed: not all languages contribute equally to the quality of results. In contrast, LSKEXTRACTOR identifies the most informative languages within clusters of similar queries, thereby reducing cost while maintaining, or even improving, performance. In Appendix A, we outline the rough runtime estimates for LSKEXTRACTOR and our baselines. The same appendix also contains extra results for the robustness of our results.

### 5 CONCLUSION

In this paper, we explore the concept of Language Specific Knowledge (LSK)—languages contain specific knowledge not present in other languages. We design a methodology, called LSKEXTRACTOR, that maps languages to specific topics. We show that LSKEXTRACTOR can improve the performance of language models by allowing them to reason in a selected language (dependent on the topic). Our extensive experimentation covers three datasets, a variety of language models (model families, parameter sizes, high-to-low resource languages), and simple to strong baselines. It shows that LSKEXTRACTOR achieves up to 10% relative improvements in accuracy, can select optimal expert languages, and is applicable in real world settings. Using the insights of this work, we hope to train models that take advantage of LSK to be more inclusive and culturally aligned.

**Future Work.** This work explored monolingual reasoning chains in language models. In the future, we will investigate: (1) multilingual reasoning chains to analyze language-switching effects on

reasoning quality, (2) more efficient methods to approximate Language-Specific Knowledge without linearly increasing computational costs, and (3) the practical impact of Language-Specific Knowledge on downstream conversational tasks like persuasion and dialogue-state tracking.

# 6 ETHICS STATEMENT

We are committed to the transparency and reproducibility of our research. We encourage our research community to make use of our open-source code to further improve our methodology. Our research involves the alignment (and potential risks that come with misalignment) in LLMs. In this work, we study this phenomenon of LSK in a controlled environment with little to no safety risks and implications. However, future work must consider these safety risks, especially in multilingual settings. Finally, we've accredited all the resources used in this paper (models, datasets, previous works), and a description of the licenses are provided in Appendix J.

# 7 REPRODUCIBILITY

For reproducibility, we not only provide detailed experimental details in Section 4, we also provide the (anonymized) code case in the abstract, and a description of the compute used during this project in Appendix I.

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

# A   OTHER RESULTS

In this section, we provide two results:

1. The robustness of LSK
2. A cost-to-performance analysis of LSKEXTRACTOR and other baselines.

## A.1   ROBUSTNESS

To show the robustness of our results, we rerun our experiments on one setting from our evaluation: CultureAtlas on Llama-8B-Instruct. Here are the results:

| Baseline | Reported above | Rerun experiment |
|---|---|---|
| OnlyEnglish | 16.37 | 16.26 |
| LLMSelected | 16.19 | 16.25 |
| GlobalLanguage | 18.87 | 18.88 |
| Majority | 17.12 | 16.06 |
| Country | 22.82 | 22.82 |
| LSKExtractor | 29.41 | 29.03 |
| LSKExtractor-top3 | 26.94 | 23.61 |

These show that our method is robust. This is because we use a temperature of 0 during all offline/online LLM inference.

## A.2   COST-TO-PERFORMANCE ANALYSIS

Furthermore, we also provide a runtime analysis of all the baselines for both offline and online costs. These results were obtained using a single A40 NVIDIA GPU. Note: all "LLM inference on train set" were done with all 16 languages. Additionally, we denote the cost for translation per language (using GPT-4o-mini) on the train set and test set as $T_{train}$ and $T_{test}$, respectively. Because different translation methods (and/or models) might take a different amount of time, we refer to the translation time as $T_{train}$ and $T_{test}$, instead of the actual time. The results are in Table 1

# B   CLUSTER ANALYSIS

Figures 7 to 9 contain the distributions of languages selected by LSKEXTRACTOR for a variety of clustering methods. They mostly show similar patterns to Figure 5 where BLEnD and Social IQa are clustered in mostly English, while CultureAtlas is clustered in a variety of languages.

We also perform a semantic topic cluster analysis in Table 2 for 12 clusters. Paired with Figure 5, we see that for the same cluster topic, each language model chooses different language experts. For example, for Cluster #5 of Culture Atlas clusters queries from Chinese customs, while the languages selected by the LLMs are Italian, Portuguese, Russian, and Chinese.

| | Offline | | | Online | | Performance |
|---|---|---|---|---|---|---|
| Baseline | Description | Cost | | Description | Cost | Accuracy |
| OnlyEnglish | None | 0 | | LLM inference on test set (one language) | 5m49s | 16.37 |
| LLMSelected | None | 0 | | LLM inference on (1) selecting a language and (2) test set | $12m34s + T_{test}$ | 16.19 |
| GlobalLanguage | LLM inference on train set | $5h3m + 16T_{train}$ | | LLM inference on test set | $5m49s + T_{test}$ | 18.87 |
| Majority | None | 0 | | LLM inference on test set (all languages) | $1h33m + 16T_{test}$ | 17.12 |
| Country | Gathering country-to-language map | Varies | | LLM inference on test set | $5m49s + T_{test}$ | 22.82 |
| LSKExtractor-top3 | LLM inference on train set + $T_{train}$ + clustering | $5h8m + 16T_{train}$ | | LLM inference on test set + finding the cluster | $6m24s + T_{test}$ | 26.94 |
| LSKExtractor | LLM inference on train set + $T_{train}$ + clustering | $5h8m + 16T_{train}$ | | LLM inference on test set + finding the cluster | $6m16s + T_{test}$ | 29.41 |

Table 1: The cost-to-performance tradeoffs for LSKEXTRACTOR and other baselines. These results show us that LSKEXTRACTOR is in the sweet spot of the cost-vs-performance tradeoff.

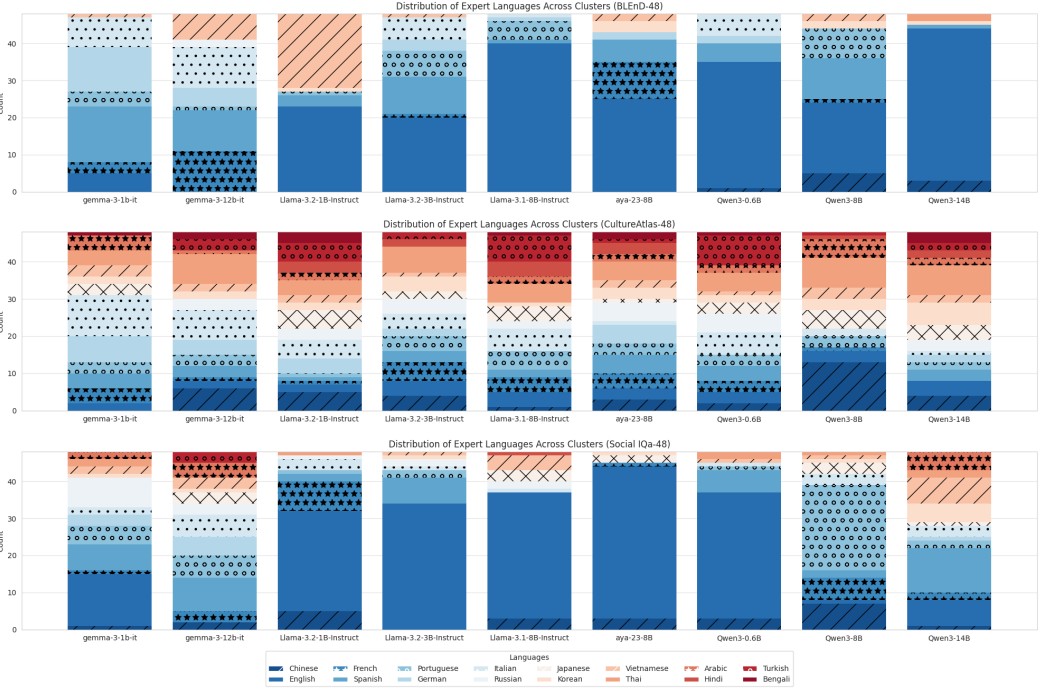

Figure 7: Distribution of languages selected across clusters (k-means with 48 clusters) for the various datasets.

| Cluster | Blend Theme | Culture Atlas Theme | Social IQA Theme |
|---------|-------------|---------------------|------------------|
| 1 | Regional specialties & industries (livestock, agriculture, tourism) | Eastern/Central European countries (Ukraine, Serbia, Czech Rep., etc.) | Basic daily activities & routine behaviors |
| 2 | Commercial hubs & popular destinations (business centers, vacation spots) | Western countries (France, Canada, Ireland, Sweden) | Social interactions & interpersonal dynamics |
| 3 | Cultural celebrations & traditional items (festivals, alcohol, ceremonies) | United States (exclusively) | Helping behaviors & consideration for others |
| 4 | Economic centers & basic needs (manufacturing, breakfast, commercial hubs) | Sub-Saharan African countries (Botswana, Niger, Ghana, etc.) | Goal-oriented actions & planning |
| 5 | Specific regional activities & entertainment (mining, music, sports teams) | China (exclusively) | Personal interests & character traits |
| 6 | Daily consumption & rivalry (food, sports rivalries, quick meals) | Middle Eastern & Mediterranean countries (Bahrain, Saudi Arabia, Greece, etc.) | Intimate relationships & emotional connections |
| 7 | Sports achievements & food origins (international success, global foods) | Southeast Asian & Island nations (Philippines, Indonesia, Malaysia, etc.) | Authority, responsibility & institutional roles |
| 8 | Family traditions & regional preferences (weekend meals, skiing, literature) | South Asian countries (India, Bangladesh, Nepal) | Complex social situations & problem-solving |
| 9 | Cultural landmarks & formal occasions (historic sites, weddings, literature) | East Asian countries (Japan, with some Fiji) | Social dynamics & behavioral expectations |
| 10 | Famous personalities & cultural celebrations (athletes, entrepreneurs, fireworks) | Oceania (Australia, New Zealand, Papua New Guinea) | Material generosity & preparation activities |
| 11 | Tourism & technology hubs (attractions, sports, tech centers) | Southeast Asian countries (Thailand, Myanmar, Cambodia, Laos) | Professional care & assistance behaviors |
| 12 | Competitive activities & popular culture (sports teams, job markets, food) | Latin American & Iberian countries (Mexico, Brazil, Spain, Peru) | Goal achievement & recreational activities |

Table 2: Cluster themes for 12 clusters across datasets. This corresponds to the results in Figures 3-5.

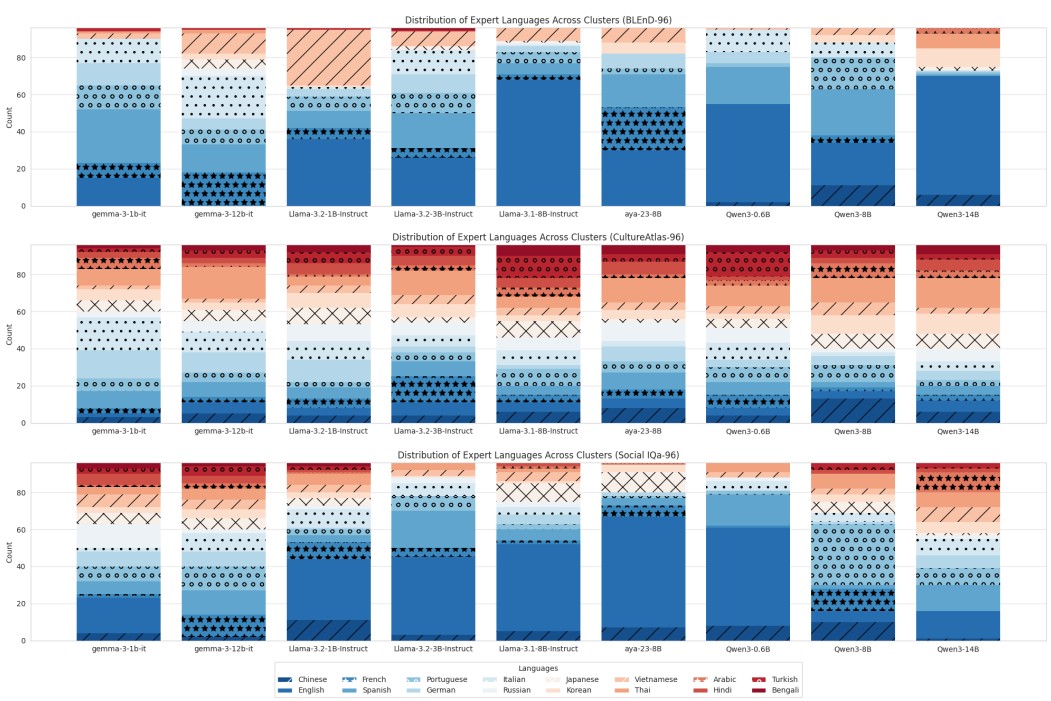

Figure 8: Distribution of languages selected across clusters (k-means with 96 clusters) for the various datasets.

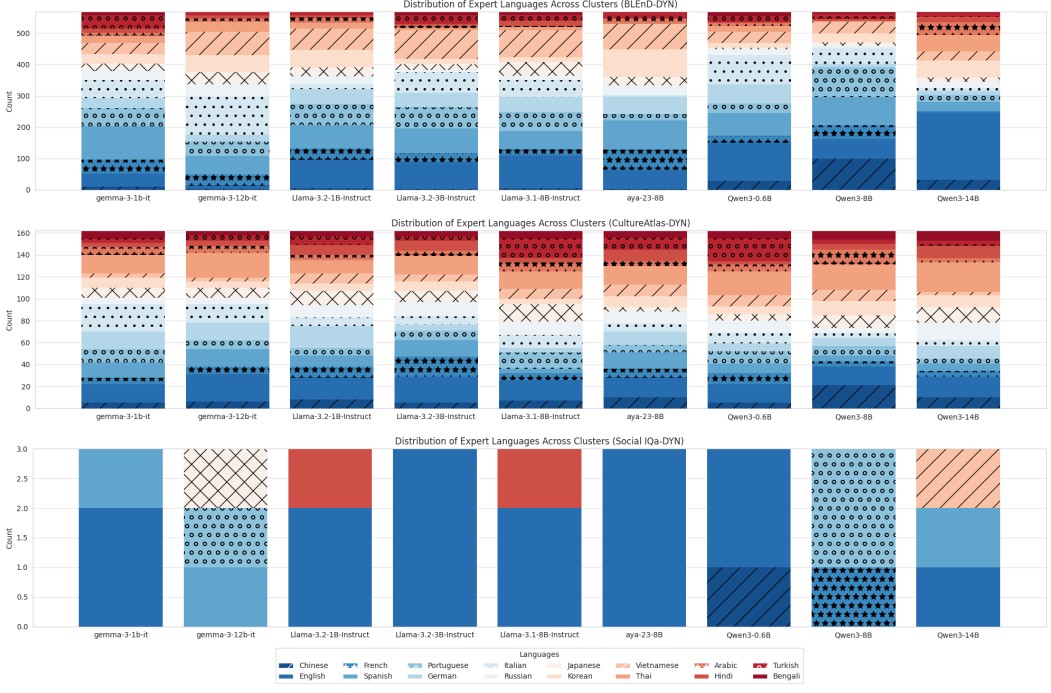

Figure 9: Distribution of languages selected across clusters (HDBSCAN) for the various datasets (BLEnD had 162 clusters, CultureAtlas had 568, and SocialIQa had 3).

## C IMPACT OF REASONING

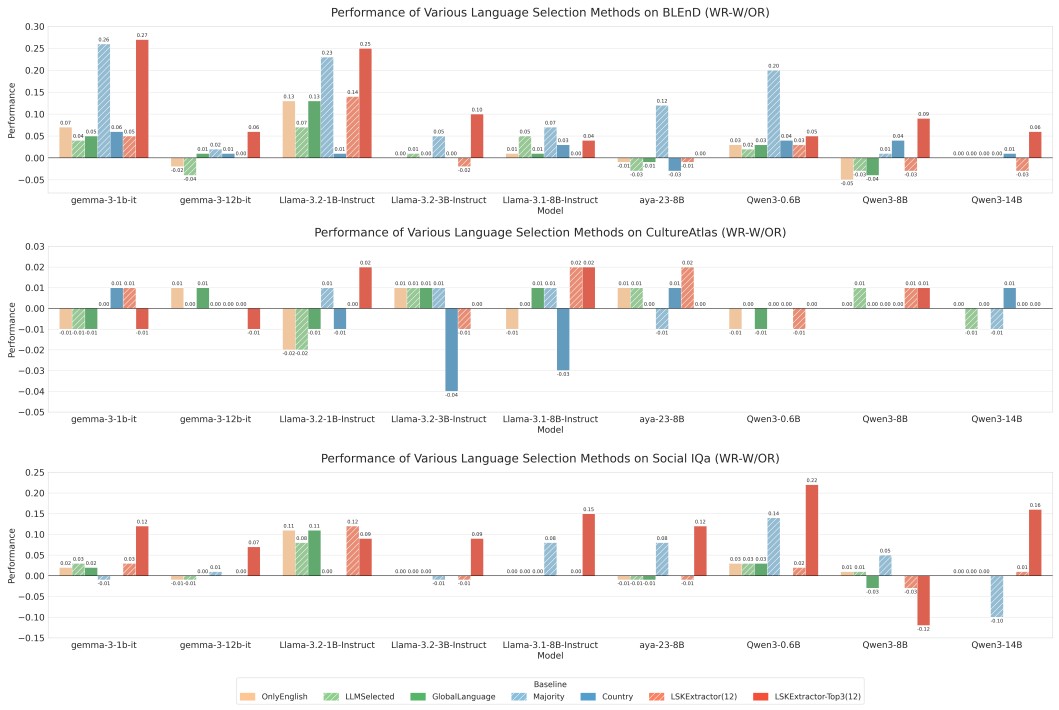

Figure 10: Measuring how important reasoning is for each model-dataset setting. These plots show the results for the difference between the performance with reasoning and without.

Figure 10 illustrates the effect of enabling reasoning by plotting the difference in performance with and without reasoning. A positive value indicates that reasoning improves accuracy, while a negative value indicates degradation. The results vary across models and datasets, but a clear pattern emerges: **smaller models tend to benefit more from reasoning than larger ones**. This trend is intuitive— larger models encode more factual and contextual world knowledge directly in their parameters and can often retrieve relevant information without additional reasoning steps. By contrast, smaller models rely more heavily on explicit reasoning to bridge knowledge gaps and organize retrieved information, leading to larger performance gains when reasoning is enabled.

| Language | Avg. Score |
|:---:|:---:|
| Hindi | 3.83 |
| Spanish | 3.93 |
| Chinese | 3.70 |
| Turkish | 3.87 |
| Portuguese | 3.60 |
| Arabic | 2.83 |
| Russian | 2.87 |
| Korean | 3.63 |
| Vietnamese | 3.63 |

Table 3: Average scores (out of 4) for translation quality, on a subset of 30 samples, judged by human annotators.

## D  DATASET TRANSLATION

To ensure the integrity of our experimental design, we translate the instructions and inputs from the datasets. The (multiple choice question-answering) datasets we choose are in English. We use OpenAI's GPT-4o-mini to translate the queries (instruction + input + answer choices). Our prompt is outlined in Figure 11.

In order to check GPT's translation quality, we ask humans to verify the translation quality. On a subset of 10 samples per dataset (totaling 30 samples), we asked participants to rate the quality of the translation from 1 (nonsense translation) to 4 (perfect translation). The average rating, broken down by language, is in Table 3.

In order to verify whether the models are outputting responses that align to the language they are supposed to reason in, we run a language classification model (specifically, `qanastek/51-languages-classifier` – we choose this for its good performance, and because it covers the language set we choose for our experimentation) and calculate the percentage of samples that follow the intended reasoning language. For BLEnD, CultureAtlas, and Social IQa, respectively, we see average accuracies of 96.97%, 97.73% and 97.93%, across all models and languages. This indicates that models generally are very good at following instructions to think in a certain language, and further strengthens the claims we make in our paper.

---

**Dataset Translation Prompt to GPT-4o-mini**

Translate ONLY the following question into {language}: "{input}".
ONLY output the translation in the following JSON format:

```
{
    "{language}_translation": <output the translated input
        here>.
}
```

---

Figure 11: Prompt to GPT-4o-mini to translate the datasets into one of the 16 languages we chose for our experimentation. As input, the translation "language" and the text to translate ("input") is provided.

# E  LLMSELECTED BASELINE DETAILS

Figure 12 contains the distribution of languages selected for the LLMSelected baseline (again, the prompt is outlined in Figure 13). As mentioned in the main text, due to English being chosen more often than LSKEXTRACTOR, LLMSelected highlights that a language model's internal LSK map is not reliable, and should be explicitly mapped.

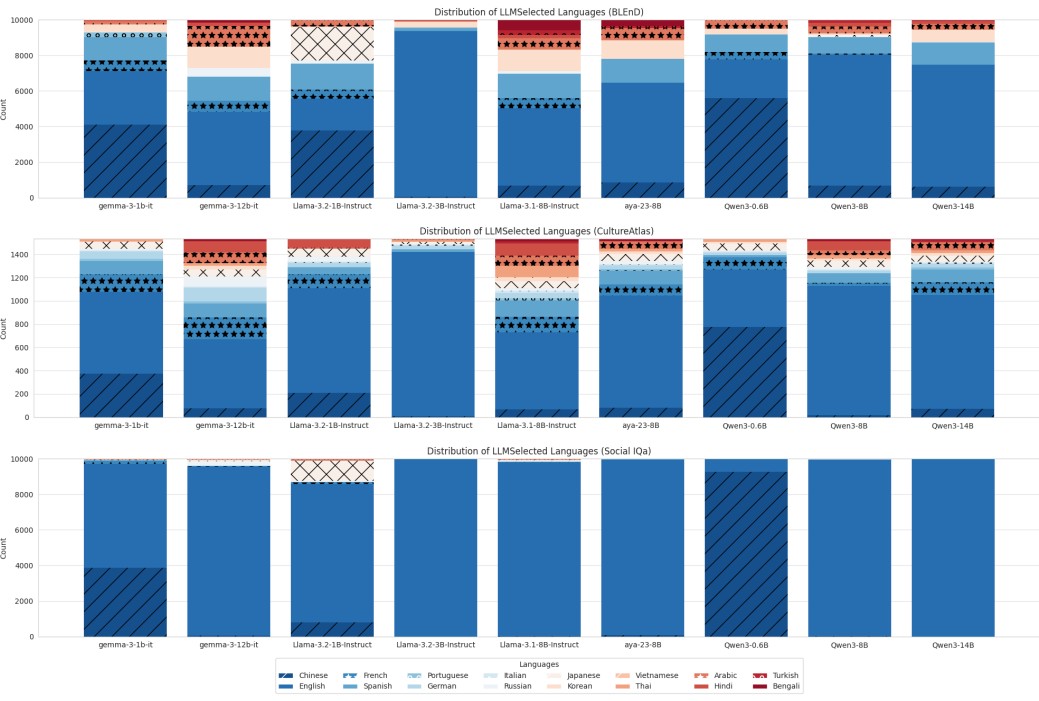

Figure 12: Distribution of languages selected by the LLMSelected baseline for each dataset.

---

**Prompt to LLM for selecting a language to best answer the question in (LLMSelected baseline)**

An expert language is the language from the provided list that is most appropriate and informative for answering the given question (e.g., because the question is about a culture, region, or source where that language is dominant, or because that language has the richest knowledge base for the topic).

From the following languages:
[Chinese, English, French, Spanish, Portuguese, German, Italian, Russian, Japanese, Korean, Vietnamese, Thai, Arabic, Hindi, Turkish, Bengali]
, determine which one is the best expert language for answering the question below.

Question: {input_question}
Fill out your language expert in the below JSON format:

```
{
  "expert_language": "<the expert language from the above list>"
}
```

---

Figure 13: Prompt to the language model to select the language expert for a given question, which is our baseline called LLMSelected.

# F  CULTUREATLAS REFORMATTING

The CultureAtlas dataset consists of cultural claims associated with specific countries, each annotated as either true or false. Because this binary classification setting is relatively simple and the dataset is imbalanced toward false claims, we reformatted it into a multiple-choice question (MCQ) format. In the reformatted version, each question presents four answer choices pertaining to the same country: one true claim and three false claims. The model is then tasked with identifying the true claim, transforming the problem into a more nuanced and challenging task that requires reasoning across all options. An illustrative example of a reformatted question is shown in Figure 14.

---

**An Example MCQ Generated from Culture Atlas**

Question: What is true about Samoa?

Answer Choices:
A. There are several different kinds of possible group structures in Samoan culture.
B. Violent crime is limited, but increasing, and public perception associates this with returns of ethnic Tongans who have been raised overseas.
C. There is no social stigma on being in prison (although that may change now too), but then of course it also does not serve as a deterrent against crimes.
D. On all other social occasions, the taualuga is usually the last dance to be performed.

Ground Truth Answer: A.

---

Figure 14: An example of a reformatted CultureAtlas question. The original binary (true/false) claims are transformed into a multiple-choice format with four options about the same country: one true claim and three false claims. The model is required to select the true claim.

## G  COUNTRY TO LANGUAGE MAPPING

For our Country Mapping baseline, we assign a language $\ell_i \in \mathcal{L}$ to each country in the dataset. For each country, we select the most commonly spoken language in the corresponding region. If the most common language is not included in $\mathcal{L}$, we default to English. The mappings used in our experiments are summarized in Table 4.

| Dataset | Language | Countries |
|---|---|---|
| **Blend** | **Arabic** | Algeria, Ethiopia |
| | **Chinese** | China |
| | **English** | Assam, Azerbaijan, Greece, Indonesia, Iran, Northern Nigeria, UK, US, West Java |
| | **Korean** | North Korea, South Korea |
| | **Spanish** | Mexico, Spain |
| **CultureAtlas** | **Arabic** | Algeria, Bahrain, Comoros, Egypt, Iraq, Jordan, Kuwait, Lebanon, Libya, Mauritania, Morocco, Oman, Qatar, Saudi Arabia, Sudan, Tunisia, United Arab Emirates, Yemen |
| | **Bengali** | Bangladesh |
| | **Chinese** | China |
| | **English** | Afghanistan, Albania, Andorra, Antigua and Barbuda, Armenia, Australia, Azerbaijan, Bahamas, Barbados, Belarus, Belgium, Belize, Bhutan, Bosnia and Herzegovina, Botswana, Bulgaria, Burundi, Cambodia, Canada, Croatia, Cyprus, Czechia, Denmark, Dominica, Eritrea, Estonia, Eswatini, Ethiopia, Federated States of Micronesia, Fiji, Finland, Gambia, Georgia, Ghana, Greece, Grenada, Guyana, Haiti, Hungary, Iceland, Indonesia, Ireland, Islamic Republic of Iran, Israel, Jamaica, Kazakhstan, Kenya, Kiribati, Kyrgyzstan, Lao People's Democratic Republic, Latvia, Lesotho, Liberia, Lithuania, Luxembourg, Madagascar, Malawi, Malaysia, Maldives, Malta, Marshall Islands, Mauritius, Mongolia, Montenegro, Myanmar, Namibia, Nauru, Nepal, Netherlands, New Zealand, Nigeria, North Macedonia, Norway, Pakistan, Palau, Papua New Guinea, Philippines, Poland, Republic of Moldova, Romania, Rwanda, Saint Kitts and Nevis, Saint Lucia, Saint Vincent and the Grenadines, Samoa, Serbia, Seychelles, Sierra Leone, Singapore, Slovakia, Slovenia, Solomon Islands, Somalia, South Africa, South Sudan, Sri Lanka, Suriname, Sweden, Tajikistan, Timor-Leste, Tonga, Trinidad and Tobago, Turkmenistan, Tuvalu, Uganda, Ukraine, United Kingdom of Great Britain and Northern Ireland, United Republic of Tanzania, United States of America, Uzbekistan, Vanuatu, Zambia, Zimbabwe |
| | **French** | Benin, Burkina Faso, Cameroon, Central African Republic, Chad, Congo, Côte d'Ivoire, Democratic Republic of the Congo, Djibouti, France, Gabon, Guinea, Monaco, Niger, Senegal, Togo |
| | **German** | Austria, Germany, Liechtenstein, Switzerland |
| | **Hindi** | India |
| | **Italian** | Italy, San Marino |
| | **Japanese** | Japan |
| | **Korean** | Democratic People's Republic of Korea, Republic of Korea |
| | **Portuguese** | Angola, Brazil, Guinea-Bissau, Mozambique, Portugal, São Tomé and Príncipe |
| | **Russian** | Russian Federation |
| | **Spanish** | Argentina, Bolivarian Republic of Venezuela, Chile, Colombia, Costa Rica, Cuba, Dominican Republic, Ecuador, El Salvador, Equatorial Guinea, Guatemala, Honduras, Mexico, Nicaragua, Panama, Paraguay, Peru, Plurinational State of Bolivia, Spain, Uruguay |
| | **Thai** | Thailand |
| | **Turkish** | Türkiye |
| | **Vietnamese** | Viet Nam |

Table 4: Country-to-language mappings used for the Blend and CultureAtlas datasets. Each country is assigned its most commonly spoken language, defaulting to English if the language is not present in $\mathcal{L}$.

# H  MODEL PROMPTS

Figures 15-18 contain the prompts to the language for during our evaluation, with and without reasoning, in three languages (English, French, Turkish) to save space. Figure 13 contains the prompt for the LLMSelected baseline in the main paper.

---

**Prompt to LLM without reasoning in English**

Question: {input_question}
Answer choices:
A. {choice_one}
B. {choice_two}
C. {choice_three}
D. {choice_four}

Select one of the answer choices. Fill out the following JSON:

```
{
    "final_answer": "<output answer here>"
}
```

---

Figure 15: Prompt to the language model to perform without reasoning. We show results for how no-reasoning affects the model performance in Figure 10. For BLEnD and Social IQa, the "input_-question" and "choice_x" comes from the dataset. For CultureAtlas, because we modify the dataset ourselves to make it more difficult, the input question will always be "Which is the following is true about {country}?". Details of the CutlureAtlas modification are in Appendix F.

---

**Prompt to LLM without reasoning in French**

Question: {input_question}
Options de réponse:
A. {choice_one}
B. {choice_two}
C. {choice_three}
D. {choice_four}

Sélectionnez l'une des options de réponse. Veuillez remplir le JSON suivant:

```
{
    "final_answer": "<votre réponse finale ici>"
}
```

---

Figure 16: Prompt to the language model to perform without reasoning. We show results for how no-reasoning affects the model performance in Figure 10. For BLEnD and Social IQa, the "input_-question" and "choice_x" comes from the dataset. For CultureAtlas, because we modify the dataset ourselves to make it more difficult, the input question will always be "Which is the following is true about {country}?". Details of the CutlureAtlas modification are in Appendix F.

```
Prompt to LLM (with reasoning) in English

Question: {input_question}
Answer choices:
A. {choice_one}
B. {choice_two}
C. {choice_three}
D. {choice_four}

Think about it in English, and then select one of the answer choices. Fill in the JSON below.

{
    "reasoning_in_English": "<your reasoning steps in English>",
    "final_answer": "<output answer here>"
}
```

Figure 17: Prompt to the language model to perform with reasoning, in English. Figure 3 illustrates the results using this prompt. For BLEnD and Social IQa, the "input_question" and "choice_x" comes from the dataset. For CultureAtlas, because we modify the dataset ourselves to make it more difficult, the input question will always be "Which is the following is true about {country}?". Details of the CutlureAtlas modification are in Appendix F.

```
Prompt to LLM (with reasoning) in Turkish

Soru: {input_question}
Cevap seçenekleri:
A. {choice_one}
B. {choice_two}
C. {choice_three}
D. {choice_four}

Türkçe olarak düşünün ve ardından cevap seçeneklerinden birini seçin. Aşağıdaki JSON'u doldurun.

{
    "reasoning_in_Turkish": "<Türkçe akıl yürütme adımlarınız>",
    "final_answer": "<çıktı cevabı buraya>"
}
```

Figure 18: Prompt to the language model to perform with reasoning, in Turkish. Figure 3 illustrates the results using this prompt. For BLEnD and Social IQa, the "input_question" and "choice_x" comes from the dataset. For CultureAtlas, because we modify the dataset ourselves to make it more difficult, the input question will always be "Which is the following is true about {country}?". Details of the CutlureAtlas modification are in Appendix F.

## I  EXPERIMENTAL SPECIFICATIONS

We run our inference on NVIDIA A40 GPUs. For the the 1B, 3B, 8B models, we used a single A40 GPU, while the 12B and 14B required two A40 GPUs. Inference takes around 30-60 minutes per language. Clustering is computationally inexpensive and can be done on a single A40 GPU.

## J  LICENSES

Our code is released publicly under the Apache-2.0 License. CultureAtlas (Fung et al., 2024) is released under the MIT License; BLEnD (Myung et al., 2025) under the CC-by-SA-4.0 License; SocialIQa (Sap et al., 2019) is not under explicit license, however it is publicly available on Hugging-face, and we do not use it for commercial purposes. All models are under their proprietary licenses from the corresponding companies.

## K  USE OF LARGE LANGUAGE MODELS

Other than being used as part of the experiments conducted in this work, LLMs were used solely as a writing assistance tool in preparing this paper submission. Their role was limited to polishing language, improving clarity, and reducing redundancy. The prompt used for this purpose was similar to "Please revise the writing of this, making sure to remove any grammatical mistakes." All research ideas, experimental designs, analyses, and claims presented in the paper are entirely the original work of the authors. No part of the conceptual, methodological, or empirical contributions relies on or originates from LLM outputs.

