# OpenReview forum: "Language Specific Knowledge: Do Models Know Better in X than in English?"
_ICLR.cc/2026/Conference — Submitted to ICLR 2026_

### Official Review · Reviewer_SyK3 · 2025-10-27

**Soundness:** 2
**Presentation:** 4
**Contribution:** 2
**Rating:** 2
**Confidence:** 4

**Summary:**

# High-level summary

The authors tackle the problem of LLMs having specific knowledge / better capabilities in specific languages, which they term as Language Specific Knowledge (LSK). LSK is derived by claiming that LLMs perform better on question-answering tasks (QA) in specific languages (like English) as opposed to other languages.

# Contributions

Their proposed contributions surrounding LSK has two parts - given a set of candidate languages:

## Part 1 (Train-time):

i) the query $Q^{train}$ is embedded and clustered using kmeans; ii) for each query in each cluster and each language $l \in L$, accuracy of the LLM is calculated and mean accuracy is calculated for the cluster per language; iii) Best language per cluster is determined.

## Part 2(Test-time):

Each test query is assigned the nearest cluster, and the language associated with that cluster from train time is used to get the final answer.

# Results

The authors experiment on three datasets: CultureAtlas, BLEnD, and Social IQa, all being/reformatted to match Multiple QA style benchmark setting and report classification accuracy.

1. There is a clear result in the claim that there is/are languages that perform better from Figure 3 and other previous works cited in the paper.
2. From Figure 3, it seems like Qwen3's family of models benefits the most from the proposed methods, other models less so (hypothesis: RL-post-trained family of models differing from the others)
3. All models are poorly self-calibrated in general to know which language they know best.
4. Distribution of the selected languages seems uniform in two datasets, but the third dataset (CultureAtlas) has wildly different distributions (which is an interesting result).

**Strengths:**

# Strengths

1. The paper is clearly written and easy to follow. Props for providing the code.
2. Clear experiments and ablations - the baselines seem well designed and performed.

**Weaknesses:**

# Weaknesses

1. The approach doesn't seem too different from [1] (cited in the paper). In [1], the authors train to classify whether a given task config and other runtime configs would work well by creating an embedding and show similar robustness to OOD tasks. Similarly, in this work, the authors first assign the query to some cluster whose best language is already known. Arguably, adding a new language here requires recomputing accuracies, as opposed to one that can be learnt online. Another visible difference is that the former method in [1] supports a lot of languages at query time, which this method would seem to require another mapping iteration. Lack of discussion comparing clear improvements, differences, and advantages/disadvantages seems critical.
2. No hyperparameters, confidence intervals/error bars, or sensitivities of the result mentioned anywhere. For example, the robustness result can also be inferred to mean that multiple language distributions are very closely related, where any choice would have yielded similar results, which would indicate a one-to-many mapping for the right set of configurations. Similarly, the impact of temperature seems critical which is not mentioned anywhere. Lack of these details and analysis surrounding them makes the observations questionable.
3. Control for thinking - in models like Qwen3, the _thinking_ mostly happens in English / Chinese; it is unclear how the authors control for thinking language/behavior and what the effect of language is in these models when thinking is enabled / disabled.
4. performance cost - another missing metric is the training cost to perceived benefit. Assuming each input and output token has a constant cost, from Figure 3, it appears that calibrating the LLMs to choose the best language would yield the best cost-to-performance ratio. There is a paragraph on Pg. 9 that very briefly discusses this, but it seems like this warrants its own specific subsection in the main content/appendix.


[1] S. Kumar, V. Balloli, M. Ranjit, K. Ahuja, S. Sitaram, K. Bali, T. Ganu, and A. Nambi. Bridging the language gap: Dynamic learning strategies for improving multilingual performance in llms. In O. Rambow, L. Wanner, M. Apidianaki, H. Al-Khalifa, B. D. Eugenio, and S. Schockaert, editors, Proceedings of the 31st International Conference on Computational Linguistics, page 9209–9223, Abu Dhabi, UAE, Jan. 2025. Association for Computational Linguistics. URLhttps://aclanthology.org/2025.coling-main.619/.

**Questions:**

same as the weaknesses. I'm willing to update the score based on answers / clarifications to the weaknesses.

---

> ### Author Response · Authors · 2025-11-20
>
> Thank you for the insightful comments and for noting that our paper is “clearly written” and that our experiments are “well designed and performed.” We address your points below.
>
> ---
>
> ### **W1. Relation to prior work and novelty**
>
> Our method for constructing the LSK map is indeed related to the assumption that similar queries benefit from similar configurations. The novelty of our work is two-fold:
> (1) we are the first to show that LSK exists for objective, knowledge-based QA tasks, and
> (2) we propose a simple and practical methodology (LSKExtractor) that allows one to utilize LSK to improve model performance.
>
> In addition, our method can be easily extended to new languages. Once inference is performed on the new language, the resulting data points are assigned to the existing clusters. Since the clusters are already defined, we only need to relabel them using the information provided by the new language. We will add these details in the camera-ready version.
>
> ---
>
> ### **W2. Hyperparameters and sensitivities**
>
> Running these multilingual experiments is computationally expensive. However, we set our sampling temperature to 0, and we released all prompts and translation details to support reproducibility.
>
> Regarding hyperparameters:
> - We already include an analysis of cluster sizes (12, 48, 96, and dynamic) in the paper.
> - Our code contains all inference hyperparameters: temperature = 0.0, top-p = 0.9, and all other settings are default.
>
> We will make these hyperparameters explicit in the camera-ready version.
>
> ---
>
> ### **W3. Controlling for thinking-language behavior**
>
> We do not use `<think></think>` tokens. Instead, as shown in Appendix G, we rely on prompting the model to output its reasoning before the answer. This avoids tying our method to specific thinking mechanisms and ensures applicability to non-thinking models as well.
>
> Thus, the effect of “thinking language” does not drive the behavior of LSKExtractor. We report the difference in results of reasoning and no reasoning in Appendix B.
>
> ---
>
> ### **W4. Cost versus performance**
>
> There is indeed a cost–performance tradeoff. LSKExtractor strikes a strong balance compared to the baselines:
>
> - Only English: cheapest, but not very performant.
> - LLMSelected: two inference calls per query, but no meaningful performance gain.
> - GlobalLanguage: similar cost to LSKExtractor (minus clustering), but weaker performance.
> - Country: inexpensive, but relies on a country-to-language mapping and is less reliable.
> - Majority: most expensive (requires inference in all languages); LSKExtractor matches or nearly matches its performance with far lower cost.
>
> We will clarify this comparison more explicitly.
>
> ---
>
> Thank you again for your careful and constructive feedback. Please let us know if you have any other questions or concerns. If you believe the revisions and clarifications above address your concerns, we would greatly appreciate seeing this reflected in an increase in scores.

---

> > ### Comment · Reviewer_SyK3 · 2025-11-23
> >
> > Thanks for the clarifications. However, I find a lot of my questions/comments are unaddressed:
> > 1. W1. The relation to prior work and novelty doesn't address/answer the novelty concerns beyond the paper that is cited and brought up in my review. While the LSK part might be novel, there is no acknowledgement/refutation of the methods being pretty similar and how the proposed method is better suited/designed in any/all scenarios.
> > 2. To clarify, in reference to W1, cost of re-labelling is $\mathcal{O}(\vert Q_{train} \vert)$ and general cost of the algorithm being $ \mathcal{O}(iters * K * \vert Q_{train} \vert * d + \vert Q_{train} * L) $ - indicating it is pretty cheap to add a new language but does not support online addition.
> > 3. This opens up a new discussion where increasing cluster sizes leads to a decrease in performance, indicating that the LSK hypothesis is only valid until a few clusters, and any increase in clusters after that results in a drop in performance, indicating LSK is no longer valid. Discussion around this seems to be missing.
> > 4. W2. Again, I'm not sure if the authors agree or disagree with me on the comment regarding:
> > >the robustness result can also be inferred to mean that multiple language distributions are very closely related, where any choice would have yielded similar results, which would indicate a one-to-many mapping for the right set of configurations
> > 5. Similarly, while I understand it is expensive to re-run all experiments with different hyperparameters, it is also crucial that the authors take a small subset, single model and analyse the difference in performance with respect to clustering, consistency of language to the cluster, etc. to provide a better context to the readers on how hyperparameters impact the method.
> > 6. W3: Is there an explicit check to ensure no other language tokens are present?
> > 7. The cost vs performance rebuttal is weak - can the authors provide rough numbers for this based on existing results and some assumptions (this does not require re-running anything, but rather just doing a token count on existing logs and computing approximate costs based on some simple assumptions and heuristics)

---

> ### Author Response · Authors · 2025-11-24
>
> ### **Comment 1: Relation to previous work**
> As mentioned in our response, yes, LSKExtractor relies on the assumption that similar queries benefit from similar configurations, as the paper the reviewer mentioned, and other fields such as AutoML [1] and graph machine learning [2]. We do not claim this assumption as our novelty.
>
> Our contribution lies instead in how we apply this assumption: we show that semantically similar content shares representations in one (or several) languages, but not necessarily across all languages. We hope this clarification helps distinguish our work from prior studies.
>
> ---
>
> ### **Comment 2: Our method can be extended to add new languages automatically**
> To make it online, we would require online user data as well (the accuracy of the LLM’s response). We can maintain a “leaderboard” of languages for each cluster and update the expert language and cluster centers as more online data is collected.
>
> ---
>
> ### **Comment 3: Cluster size’s effect on performance**
> We choose a relatively small subset of languages and a contained dataset, which might not have many topics. But real-world data would have a lot more topics that contain LSK: religion, history, legal literature, global literature, art, etc. Our work serves as a proof of concept that (1) LSK exists and (2) LSK can be leveraged. We will also include a discussion on the effects of cluster sizes in the final version of our paper.
>
> ---
>
> ### **Comment 4: Language distributions are closely related**
> While multiple languages may perform well within a given cluster, there is often still a leading language. For example, in Cluster 5 of SocialIQa with Qwen-8B, the top three languages are Spanish (71.7%), Chinese (71.7%), and English (71.1%). This suggests that a cluster may map to a small set of expert languages rather than a single one. However, we also observe counterexamples. In Cluster 10 of SocialIQa with Llama-1B, the top three languages are English (64.5%), French (50.9%), and Portuguese (49.2%), which show a much larger gap. LSKExtractor automatically handles these variations in language dominance across clusters.
>
> ---
>
> ### **Comment 5: Robustness results on a subset**
> Sure, we can rerun experiments on a subset and report back the results! We’d appreciate the reviewer’s patience while we run those experiments!
> ---
>
> ### **Comment 6: Verification of language-specific reasoning**
> Yes, we verify that the models were outputting reasoning in the language they were specified to reason in. Please check line 1082 (Appendix C). We run the LLM outputs through a language classifier and find that around 97% of the LLM outputs were in the correct language.
>
> ---
>
> ### **Comment 7: Cost vs Performance**
> While we are rerunning the experiments on a subset of data, we can also get exact numbers for the cost of each baseline. Again, we’d appreciate the reviewer’s patience!
>
> ---
>
> Thank you for the comments and questions. We will update you when those experiments are finished running. In the meantime, please let us know if you have any more questions and suggestions.
>
> [1] Koren, O., Hallin, C. A., Koren, M., & Issa, A. A. (2022). AutoML classifier clustering procedure. International Journal of Intelligent Systems, 37(7), 4214-4232.
>
> [2] Chu, Y. W., Tenorio, E., Cruz, L., Douglas, K., Lan, A. S., & Brinton, C. G. (2021, December). Click-based student performance prediction: A clustering guided meta-learning approach. In 2021 ieee international conference on big data (big data) (pp. 1389-1398). IEEE.

---

> > ### Author Response · Authors · 2025-11-27
> >
> > We appreciate Reviewer SyK3's patience while we run our experiments. We have addressed your questions in the global comment. Please let us know if you have any further questions or comments!

---

### Official Review · Reviewer_8FzN · 2025-11-02

**Soundness:** 2
**Presentation:** 2
**Contribution:** 2
**Rating:** 4
**Confidence:** 4

**Summary:**

This study propose a method to QA task - measuring the ability of the LLMs by changing the language of the input query.
It introduces the teram "Language Specific Knowledge (LSK)" to denote question that are best answered in an *expert* language for an LLM.

**Strengths:**

The paper formalizes "Language Specific Knowledge (LSK)"" that some queries are better answered when the model reasons in a particular "expert language" and turns it into a concrete language-selection problem. The proposed two-stage LSKEXTRACTOR (map LSK via clustering and pick expert language at inference) is simple and well-motivated.

**Weaknesses:**

- While two step-process is well-motivated, however, it is not very much clear why clustering was needed as in these datasets QAs are language specific already. Is it that find to create/find a cluster that represent an expert for a language?
- The CultureAtlas binary -> MCQ reformulation is sensible how distractors/other options are generated is not clear.
- Queries/questions are translated, however, not manual verification (even on small samples) is provided.

Typos/Grammatical/Minor issues
- The math notation for CoT reasoning is not clear, it is conditioned with l. Any other notation can be used to define CoT.
- L187: I assume queries also belongs to l.

**Questions:**

- L150: What was the reason for translation? As mentioned in the weaknesses, datasets are already language specific, therefore, it is not clear why translation was needed?
- L184: What is the reason to embed the English version only?
- - What is the reason for majority voting?

---

> ### Author Response · Authors · 2025-11-20
>
> Thank you for the suggestions and for noting that LSKExtractor is “simple and well-motivated.” Below, we address the comments and questions.
>
> ---
>
> ### **W1. Why clustering is needed**
>
> There seems to be a slight misunderstanding: the datasets are not inherently language-specific. BLEnD and CultureAtlas include country metadata, but Social IQa does not, and all datasets are originally in English.
>
> Since LSK assumes that models know more about certain topics in certain languages, clustering enables us to discover these topics, i.e., groups of semantically similar queries. After clustering, we assign each cluster an expert language based on performance. Appendix A (Table 1) contains examples of the cluster categories produced by LSKExtractor.
>
> ---
>
> ### **W2. CultureAtlas reformulation details**
>
> We expand on this in Appendix E. To clarify here:
>
> CultureAtlas is composed of True/False questions. We convert them into multiple-choice by grouping four binary claims into one MCQ. For example:
>
>     Which of the following is true about `<country_name>`?
>     A. C1
>     B. C2
>     C. C3
>     D. C4
>
> where only `C3` is true.
>
> We will clarify this procedure in the final version of the paper.
>
> ---
>
> ### **W3. Human verification of translations**
>
> We performed human verification of GPT-generated translations. Participants rated translation quality on a 1–4 Likert scale:
>
> - 1 = nonsense
> - 2 = major errors but understandable
> - 3 = minor errors, semantics preserved
> - 4 = perfect translation
>
> We sampled 10 items from each dataset (30 total), and reported the average score for all 30 samples. Results:
>
> | Language     | Avg. Score |
> |--------------|------------|
> | Hindi   | 3.83 |
> | Spanish | 3.93 |
> | Chinese | 3.70 |
> | Turkish | 3.87 |
> | Portuguese | 3.60 |
> | Arabic  | 2.83 |
> | Russian | 2.87 |
>
> We are still collecting additional human evaluations and will report the updated scores.
>
> ---
>
> ### **W4. CoT notation clarity**
>
> We use “conditioned with $l$” to indicate that CoT reasoning is generated in language $l$ (via a translated “think step by step” prompt). All main experiments use CoT (except Appendix B). For clarity, we can remove the explicit conditioning in the notation and instead state that $Q_l$ prompts reasoning in language $l$.
>
> ---
>
> ### **W5. Clarifying notation for queries and clusters**
>
> Queries do not belong to a language $l$ because each query is translated into every target language. In Line 187, $Q \in C_j$ means that the embedding of $Q$ belongs to cluster $C_j$, yet no language has been assigned at this stage.
>
> ---
>
> ### **Q1. Why translation was needed**
>
> We translate the queries because all datasets are originally in English, and translation helps in two ways:
>
> 1. It strengthens the LSK signal by allowing the model to leverage language-specific syntactic and cultural priors.
> 2. Preliminary experiments showed that prompting the model in English to “think step by step in `<language>`” was unreliable; direct translation produced far more stable behavior.
>
> ---
>
> ### **Q2. Why embed only the English version**
>
> We embed queries in their original English form because this aligns with the real inference setting: a single incoming query must be embedded once and assigned to a cluster. Embedding multiple translated versions would introduce unnecessary latency.
>
> Embedding models may also encode information differently across languages; we plan to explore this in future work.
>
> ---
>
> ### **Q3. Why majority voting appears**
>
> Majority voting is used only as an upper-bound baseline in evaluation. For constructing the LSK map, we do not use majority voting in the variance-reduction sense. Instead, for each cluster, we simply select the best-performing language across the queries in that cluster.
>
> ---
>
> Thank you again for your careful and constructive feedback. Please let us know if you have any other questions or concerns. If you believe the revisions and clarifications above address your concerns, we would greatly appreciate seeing this reflected in an increase in scores.

---

### Official Review · Reviewer_akTQ · 2025-11-03

**Soundness:** 3
**Presentation:** 3
**Contribution:** 3
**Rating:** 6
**Confidence:** 3

**Summary:**

This work analyses phenomenon in which multilingual large language models process questions with identical meanings differently across languages, proposing the concept of Language-Specific Knowledge (LSK). The core assumption of LSK is that within large language model some languages contain more information about certain knowledge regions than others. To do this authors propose a method called LSKExtractor. During training, the model builds semantic embedding clusters, and during inference it uses these clusters to generate answers in the language that demonstrates best performance. The authors shows through experiments on 9 models and 3 datasets that language choice can enhance the model performance.

**Strengths:**

- A novel approach to enhance model performance by leveraging the language that yields best results rather than suppressing the use of other languages.
- The method can improve the performance in a straightforward way, without any additional training.

**Weaknesses:**

- The choice of expert language varies significantly depending on which dataset is used training the LSK map. In Figure 5, CultureAtlas results in diverse language selection, while BLEnD and Social IQa predominantly select English.
- The paper does not address how to evaluate open-ended prompts that cannot be evaluated using accuracy. In real-world scenarios prompts are far more diverse in form and intent.
- The paper focuses only on culturally grounded datasets without testing whether LSK also appears in other domains such as mathematics or coding. Adding experiments in these domains would help verify generality of the approach.

**Questions:**

- Which embedding model was used? The paper does not specify which embedding model was employed.
- Chinese is likely selected because it constitutes for a large portion of the training data, but why was Portuguese chosen as an expert language?

---

> ### Author Response · Authors · 2025-11-20
>
> Thank you for the thoughtful and constructive review, and for recognizing that our work introduces *“a novel approach”* that improves multilingual model performance *“without any additional training.”* We address each of your comments below.
>
> ---
>
> ### **W1. Variation in expert-language selection across datasets**
>
> We appreciate this observation and agree it highlights a central strength of LSKExtractor.
> CultureAtlas contains highly culture-specific knowledge, so different languages encode different regions of knowledge, resulting in diverse expert-language assignments. In contrast, BLEnD and Social IQa involve less culturally grounded knowledge, where English is assumed to be the best-performing, expert language. Even in these settings, LSKExtractor still identifies non-English expert languages for certain clusters and consistently matches strong baselines. This demonstrates that LSKExtractor adapts to different datasets while maintaining top performance.
>
> ---
>
> ### **W2. Applicability beyond accuracy-based evaluation**
>
> We agree that open-ended generation tasks require different evaluation criteria. We could easily extend our method to generation tasks by thresholding a performance metric. For example, if we are creating an LSK map for a summarization task and we use an n-gram similarity metric for accuracy, we could apply a threshold to determine good and bad performance. We will clarify this extension in the camera-ready version.
>
> ---
>
> ### **W3. Generality beyond culturally grounded datasets**
>
> Our current scope focuses on datasets where language-specific knowledge is most likely to manifest. Domains such as mathematics or programming are primarily represented in English in pretraining corpora, and indeed, we expect little LSK there. Social IQa already serves as a partially culture-agnostic dataset, where we show that LSK exists, but not as prevalent as other datasets. Still, LSKExtractor is not affected by this and maintains performance.
>
> ---
>
> ### **Q1. Embedding model specification**
>
> Thank you for pointing this out. We used the **`Qwen/Qwen3-Embedding-0.6B`** model due to its strong MTEB performance and will add this detail to the final version.
>
> ---
>
> ### **Q2. Why Portuguese emerges as an expert language**
>
> Portuguese was selected because, for that particular cluster, it achieved the highest measured accuracy. As you note, some expert-language assignments may appear unintuitive. This is precisely why methods based solely on heuristic or rule-based language selection may overlook useful latent knowledge encoded in multilingual LLMs. LSKExtractor, by contrast, directly surfaces these patterns and can provide insight into pretraining data distributions of closed-source models. We will clarify this interpretation in the revision.
>
> ---
>
> Thank you again for your careful and constructive feedback. Please let us know if you have any other questions or concerns. If you believe the revisions and clarifications above address your concerns, we would greatly appreciate seeing this reflected in an increase in scores.

---

### Author Response · Authors · 2025-11-27

Dear reviewers, ACs, PCs, and other conference organizers,

We appreciate all the feedback, suggestions, and comments given to us so far. We’d like to provide a few extra experiments in a global comment.

### **Sensitivity of results**
To show the robustness of our results, we rerun our experiments on one setting from our evaluation: CultureAtlas on Llama-8B-Instruct. Here are the results:
| Baseline          | Reported in the paper | Rerun experiment |
|-------------------|-----------------------|------------------|
| OnlyEnglish       | 16.37                 | 16.26            |
| LLMSelected       | 16.19                 | 16.25            |
| GlobalLanguage    | 18.87                 | 18.88            |
| Majority          | 17.12                 | 16.06            |
| Country           | 22.82                 | 22.82            |
| LSKExtractor      | 29.41                 | 29.03            |
| LSKExtractor-top3 | 26.94                 | 23.61            |

These show that our method is robust. This is because we use a temperature of 0 during all offline/online LLM inference.

### **Runtime analysis**
Furthermore, we also provide a runtime analysis of all the baselines for both offline and online costs. These results were obtained using a single A40 NVIDIA GPU. Note: all “LLM inference on train set” were done with all 16 languages. Additionally, we denote the cost for translation per language (using GPT-4o-mini) on the train set and test set as $T_{train}$ and $T_{test}$, respectively. Because different translation methods (and/or models) might take a different amount of time, we refer to the translation time as $T_{train}$ and $T_{test}$, instead of the actual time.

|                   | Offline                                 |          | Online                                                     |           | Performance |
|-------------------|-----------------------------------------|----------|--|------------------------------------------------------------|-----------|
| Baseline          | Description                             | Cost     | Description                                                | Cost      | Accuracy |
| OnlyEnglish       | None                                    | 0        | LLM inference on test set (one language)                   | 5m49s     | 16.37 |
| LLMSelected       | None                                    | 0        | LLM inference on (1) selecting a language and (2) test set | 12m34s+$T_{test}$  | 16.19
| GlobalLanguage    | LLM inference on train set              | 5h3m+16$T_{train}$ | LLM inference on test set                                  | 5m49s+$T_{test}$   | 18.87
| Majority          | None              | 0 | LLM inference on test set (all languages)                  | 1h33m+16$T_{test}$ | 17.12
| Country           | Gathering country-to-language map       | Varies   | LLM inference on test set                                  | 5m49s+$T_{test}$   | 22.82
| LSKExtractor      | LLM inference on train set + $T_{train}$ + clustering | 5h8m+16$T_{train}$ | LLM inference on test set + finding the cluster            | 6m16s + $T_{test}$    | 29.41
| LSKExtractor-top3 | LLM inference on train set + $T_{train}$ + clustering | 5h8m+16$T_{train}$ | LLM inference on test set + finding the cluster            | 6m24s  + $T_{test}$   | 26.94


These results show us that LSKExtractor is in the sweet spot of the cost-vs-performance tradeoff.

---

### Author Response · Authors · 2025-12-01
**Revised draft of paper**

Dear reviewers, ACs, PCs, and other conference organizers,

We have updated our submitted paper draft with revisions from the rebuttal so far (in both the main text and appendix). Please find them highlighted in red in the draft. We are happy to incorporate more feedback and would gladly revise the draft if our work is accepted.

Thank you for all your efforts in reviewing our work!

---

### Meta-Review · Area_Chair_YRLf · 2025-12-17

**Summary:**

The paper shows a novel bias of common LLMs: they might perform better at a given QA task if prompted with a language than with another, and that the best language is not necessarily English.
While the reviewers appreciated the novelty of the finding and the clarity of the presentation, they also raised concerns about the significance and generality of the results of the paper, that were not sufficiently addressed during the rebuttal:
1. Experiments are run on three multiple-choice question datasets, and only on one dataset (CultureAtlas) the gap between the baseline "Global Best Language" and the proposed method seems large enough to justify the cost to select the best language for each query.
2. No confidence intervals nor significance analysis are provided. These would be especially important for the datasets Blend and Social IQa, and for the experiment on the sensitivity to the number of clusters, where the accuracy gaps between methods are small.

**Reviewer Concerns:**

Reviewers' concerns that were addressed include:
1. Relation to prior work and novelty.
2. Controlling for thinking-language behavior.
3. Cost vs performance.

The following major issues were not sufficiently addressed:
1. Applicability beyond accuracy-based evaluation: While the authors explained how to extend the method to non-accuracy-based tasks, experiments were not reported on additional datasets.
2. Hyperparameters, confidence intervals/error bars, or sensitivities: See summary section.
3. Human verification of translations: The newly added verification (Appendix D) does not report number nor demographics of the evaluators, instructions given, significance of the results given the size of the selected subset.

**Reviewer Scores:**

akTQ) Their score might have increased as almost all their concerns were addressed.
8FzN) I cannot express a judgement based on the available information.
SyK3) Their score might have increased to a 4.

---

### Decision · Program_Chairs · 2026-01-26

Reject